# Extracellular Vesicles as Potential Biomarkers in Amyotrophic Lateral Sclerosis

**DOI:** 10.3390/genes14020325

**Published:** 2023-01-27

**Authors:** Maruša Barbo, Metka Ravnik-Glavač

**Affiliations:** Institute of Biochemistry and Molecular Genetics, Faculty of Medicine, University of Ljubljana, SI-1000 Ljubljana, Slovenia

**Keywords:** amyotrophic lateral sclerosis, ALS, extracellular vesicles, biomarkers

## Abstract

Amyotrophic lateral sclerosis (ALS) is described as a fatal and rapidly progressive neurodegenerative disorder caused by the degeneration of upper motor neurons in the primary motor cortex and lower motor neurons of the brainstem and spinal cord. Due to ALS’s slowly progressive characteristic, which is often accompanied by other neurological comorbidities, its diagnosis remains challenging. Perturbations in vesicle-mediated transport and autophagy as well as cell-autonomous disease initiation in glutamatergic neurons have been revealed in ALS. The use of extracellular vesicles (EVs) may be key in accessing pathologically relevant tissues for ALS, as EVs can cross the blood–brain barrier and be isolated from the blood. The number and content of EVs may provide indications of the disease pathogenesis, its stage, and prognosis. In this review, we collected a recent study aiming at the identification of EVs as a biomarker of ALS with respect to the size, quantity, and content of EVs in the biological fluids of patients compared to controls.

## 1. Amyotrophic Lateral Sclerosis

Amyotrophic lateral sclerosis (ALS) is described as a fatal and rapidly progressive neurodegenerative disorder, caused by the degeneration of upper motor neurons in the primary motor cortex and lower motor neurons of the brainstem and spinal cord [1,2]. The destruction and ultimately the death of neurons connecting the brain or spinal cord to muscles results in paralysis and eventually death from respiratory failure within 3–5 years of disease onset [3]. Neurons that control mouth, throat, chest, arm, and leg muscles degenerate, resulting in weakness, spasticity, and atrophy of the associated muscles [4]. The heterogeneous disorder known for its highly variable phenotype starts either as a bulbar- or limb-onset disease or as a trunk/respiratory involvement disease that further spreads to other regions [5]. Besides the motor symptoms, which are the most typical disease indicators, the disease is associated with mild cognitive deficits in about one-quarter of ALS patients as well as with frontotemporal dementia (3–5%) [2]. With factors such as early respiratory muscle dysfunction, bulbar-onset disease, and older age of onset significantly reducing the survival of ALS patients, other factors such as limb-onset disease, younger age of onset, and diagnostic delay predict longer survival [5]. ALS has a prevalence rate of 3–5 in 100,000 people and an incidence rate of 1–2 in 100,000 people in the European population [6]. Typically occurring between the ages of 40 to 70 years, ALS appears to be more common in white males compared to females, notably in the case of the sporadic form of the disease [5,7]. Sporadic ALS (sALS) accounts for the vast majority (90%) of ALS cases, whereas the remaining 10% of cases constitute the familial ALS (fALS) type, which is inherited in an autosomal-dominant way [8]. Despite the causes of sALS remaining largely unknown, the risk factors of ALS include smoking [9,10], loss of body mass index [11,12], intense physical activity [13], exposure to chemicals, pesticides, heavy metals, and viruses [14,15,16,17,18], as well as some medical conditions namely head trauma, metabolic diseases, cancer, and neuroinflammation [2]. On the other hand, a higher intake of antioxidants such as vitamin E and polyunsaturated fatty acids was found to lower the risk of ALS [19]. Although non-genetic factors are thought to have a considerable impact on the etiology of ALS discrepancies exist, therefore, the evidence base for these risk factors is still limited [2].

## 2. Genes Involved

Aside from the abovementioned environmental factors, genetic factors and cellular dysfunctions are described to take part in the etiology of ALS. The main genes associated with ALS pathogenesis are copper- and zinc-containing antioxidant superoxide dismutase 1 (SOD1), TAR DNA-binding protein 43 (TARDBP, also known as TDP43), fused in sarcoma (FUS), optineurin (OPTN), valosin-containing protein (VCP), ubiquilin 2 (UBQLN2), C9ORF72, and profilin 1 (PFN1). However, altogether more than 30 genes reported as monogenic causes of ALS were found up to date [20]. These monogenic forms account for approximately 60–70% of fALS and only 10–15% of sALS, with the SOD1 mutation and GGGGCC hexanucleotide expansion mutation in the 5′ noncoding region of C9ORF72 representing the prevailing genetic causes [7,21]. Therefore, the remaining 85% of sALS are suggested to be multigenetic cases and/or the result of the interaction of genetic and non-genetic factors. The latter involve environmental risk factors and behavioral factors, epigenetic modification, or DNA damage [20]. Genetic and non-genetic factors are proposed to potentially cause cellular dysfunction including glutamate excitotoxicity [22]; abnormal protein aggregation [23]; prion-like propagation [24,25]; endoplasmic reticulum stress [26,27,28,29]; autophagy dysregulation [30,31,32,33]; secretion of neurotoxic vesicles by surrounding cells such as astrocytes and muscle cells [34,35]; alteration of RNA processing [36]; and mitochondrial disorganization and dysfunction leading to oxidative stress [37,38,39,40]. Those pathophysiological processes, individually or collectively, lead to the altered electrophysiological properties of the motor neuron and motor neuron stress, ultimately resulting in motor neuron death. Additionally, pathophysiological processes underlying ALS may occur not only in motor neurons but also in surrounding cells that aberrantly secrete neurotoxic elements contributing to motor neuron death and consequently participating in disease propagation [20]. In particular, activated microglia may contribute to motor neuron injury by producing and releasing neurotoxic superoxide and nitric oxide metabolites [41]. Similarly, astrocytes are selectively toxic to motor neurons due to the expression of mutant SOD1 [34,42,43,44] and secondly due to impaired clearance of extracellular glutamate caused by a defective glutamate transporter [45].

Apart from coding chromosomal regions, mutations occurring in the non-coding DNA are also involved in the progress of ALS as suggested by genome-wide association studies (GWAS) [46]. Non-coding DNA sequences take up a large part of human DNA and play an important role in cellular function. Indeed, regions of non-coding DNA such as enhancers regulate the expression of coding genes by binding transcription factors [47,48]. For instance, a study by Cooper-Knock et al. discovered ALS-associated pathogenic genetic variation within caveolin 1/caveolin 2 (CAV1/CAV2) enhancers. Identified CAV1/CAV2 mutations result in reduced gene expression and ultimately neurodegeneration [49]. Similarly, in a large-scale case-control association study conducted among Japanese ALS patients, a functional SNP was found in an enhancer region of ZNF512B at chromosome 20q13. The mutation is thought to be significantly associated with ALS progression because it results in decreased expression of ZNF512B, followed by decreased TGF-β signaling finally resulting in neuronal degradation [50]. At this point, it is worth noting that there exists phenotypic variability between patients of different ethnic backgrounds [51]. All in all, the C9ORF72 repeat expansion, which is the most common genetic abnormality in fALS as described above, is also located in a non-coding DNA region [52].

## 3. Diagnosis of ALS

Due to ALS’s slowly progressive characteristic, often accompanied by other neurological comorbidities, its diagnosis remains challenging for numerous reasons. Not only is ALS estimated to start years before the diagnosis and is usually diagnosed 10–16 months after the symptom onset, but it is also often misdiagnosed [1,53]. For instance, considerable biological changes such as structural degeneration of the brain and spinal cord have been confirmed in both SOD1 and C9ORF72 mutation carriers long before symptom manifestation [54,55].

ALS diagnosis is based on three main principles, including symptoms of malfunction of a certain body part, the manifestation of central and peripheral motor neuron signs in one or more segmental anatomical areas, and the functional impairment progression [56]. The diagnosis consists of clinical evaluation, evaluation of medical history, and physical examination, together with electromyography (EMG), neuroimaging, and laboratory diagnostics to rule out disorders that may resemble ALS [57,58]. EMG is used to confirm the extent of muscle denervation and identify ALS-mimicking diseases [59]. With the support of neuroimaging structural lesions impinging on motor tracts can be eliminated. Furthermore, genetic testing of the most common mutations such as C9ORF72, SOD1, TDP43, FUS, and TBK1 can be used. Not only in the diagnosis but also in the prognosis of ALS biomarkers are taking an important part as they can reveal the disease onset already in the initial phase before the symptoms of upper motor neuron disease [1].

## 4. Biomarkers in ALS

ALS biomarkers have been the subject of extensive research over the past two decades. Better understanding of the disease among the clinicians and researchers could lead to improved design of clinical trials and the development of novel therapeutics, ultimately resulting in improved outcomes in ALS patients. Indeed, there is a large amount of research data on the field of ALS biomarkers, yet few have in fact been validated. The reason for this lies in the variation of methodology, non-standardized analytical procedures, insufficient patient cohort, and deficit of longitudinal studies. Therefore, the exploration and validation of biomarkers that would shorten the diagnostic process and improve diagnostic accuracy extensively continues. Classification of ALS biomarkers includes diagnostic biomarkers, prognostic and predictive biomarkers, as well as pharmacodynamic and disease progression biomarkers [60].

The class of systemic prognostic biomarkers is represented by body weight measurement and measurement of respiratory function. Malnutrition and weight loss have an adverse effect on life expectancy in ALS [61]. Analogously, respiratory function reduces with the evolution of ALS or is already reduced at presentation in the respiratory onset disease [60].

Regarding circulating biomarkers, a wide variety of candidates, for instance, proteins, microRNA (miRNA), mRNA, and metabolites, have been investigated across the studies, not only in the cerebrospinal fluid (CSF) but also in the blood and urine [20]. CSF carries a variety of useful biomarkers since the biofluid is adjacent to the brain and spinal cord. The most promising candidates for CSF analysis are neurofilament proteins or so-called neurofilaments, consisting of three subunits. Two of them, namely neurofilament light chain (NfL) and phosphorylated form of neurofilament heavy chain (pNfH) have been validated as diagnostic biomarkers with an increase of either being related to a shorter survival [62,63,64,65,66]. To elaborate, their accumulation has been reported following axonal damage and degeneration, which is however not specific to ALS as it has also been associated with Alzheimer’s disease (AD) [67], X-linked adrenoleukodystrophy [68], and dementia in adult patients with Down syndrome [69]. Aside from high levels of neurofilament proteins, proteomic analyses have revealed raised complement C3 and secretogranin I, as well as reduced cystatin C and transthyretin [70,71,72]. However, decreased transthyretin and cystatin C levels are also altered in other neurodegenerative diseases (ND) such as AD, which does not make them specific to ALS patients [20]. Chitotriosidase (CHIT1) and glutamate receptor 4 were found to be significantly elevated in ALS patients [73]. The latter correlates negatively with ALS severity, implicating over-expression in the early stages of the disease, thus earlier treatment with anti-glutamate treatment, as is riluzole, is more effective [74].

Apart from proteomics, metabolomic studies have been performed on biofluids to access circulating metabolites that quantitively differ in ALS patients. Reduced CSF acetate levels and elevated pyruvate and ascorbate levels were observed when ALS patients were compared to the non-ND controls [75]. This, together with elevated levels of creatine and glucose [76], shows a general trend of the upregulation of metabolites, which is in line with the hypermetabolism that is often observed in ALS patients [77]. Other indicators of dysregulated metabolism observed in plasma and CSF include elevated levels of total cholesterol and LDL cholesterol [78,79], as well as homocysteine, a representative of the neurotoxic metabolites class [80]. 

Oxidative stress and neuroinflammation are known contributors to ALS pathologies; therefore, altered levels of misfolded SOD1 and glutathione, as well as cytokines IL-10, IL-6, GM-CSF, IL-2, and IL-15 [81], may potentially serve as biomarkers [60]. Besides the aforementioned cytokines, IL-17, bFGF, VEGF, MIP1b, MIP-1α, MCP-1β, and IFN-γ [82] as well as follistatin, IL-1α, and kallikrein-5 have been identified as biomarkers of neuroinflammation [83]. Glutathione, a marker of oxidative stress, was found significantly decreased in ALS patients compared to controls [84]. Measurement of mutated SOD1 protein levels in CSF, however, may only be useful as a pharmacodynamic biomarker, thus tracking the effects of antisense oligonucleotide (ASO) SOD1-lowering therapy. This is because there was no significant difference in protein levels between patients with and without SOD1 mutations, as well as between patients with ALS and controls [85,86]. They also identified urinary 8-oxodeoxyguanosine (8-OHdG) and 15-F(2t)-isoprostane (15-F(2t)-IsoP) [87], CSF-derived 8-hydroxy-2′-deoxyguanosine [88], and 3-nitrotyrosine [89], as well as 4-hydroxy-2,3-nonenal in both serum and CSF [90]. To date, these oxidative biomarkers have not yet been validated for use in clinical trials [60].

In addition to mutated SOD1 proteins, toxic aggregates of C9RAN dipeptides, resulting from C9ORF72 mutations, may be used as pharmacodynamic biomarkers [91]. These dipeptides are measurable in the CSF of ALS patients, but not controls, and their levels were not only found stable over time but also reduced due to ASO treatment in cell and mouse models [92].

The identification of miRNAs detected in CSF, as well as in the plasma and serum of ALS patients, revealed differentially expressed miRNAs compared to controls without the disease. Firstly, TARDBP, a hallmark of disease pathology, is involved in the processing of miRNA. As expected, dysregulation of TARDBP-binding miRNAs was reported in the CSF and serum of sALS patients [93]. Secondly, other identified dysregulations include the upregulation of miR-338-3p and miR181a-5p and the downregulation of miR21-5p and miR15b-5p [94,95].

Several circulating CSF biomarkers are also blood-based as they are transported between the two biofluids, whereas other candidate blood biomarkers result from the disease processes at the periphery; for instance, denervation changes in muscles [20]. Carriers of SOD1 mutation showed reduced SOD1 levels in leukocytes following pyrimethamine treatment [96]. Furthermore, poly(GP) proteins were found in peripheral blood mononuclear cells of C9ORF72 mutation carriers [92]. 

Although increased global DNA methylation has been detected in the blood of ALS patients, the evidence in different cell types is inconsistent and thus requires further research [20]. Regarding neurofilament proteins, blood NfL levels seem to be increased in ALS patients and remain steady over time, making them potentially useful as diagnostic and prognostic markers, whereas pNfH levels were shown to be very inconsistent. Data from studies of several immune activity blood markers reveal the increased immune activity of ALS patients. To elaborate, altered levels of T-regulatory cells were identified as promising prognostic biomarker candidates; however, studies of cytokines, CRP, and chitotriosidase show contradictory results [20]. Muscle denervation markers include lower serum creatinine [87], as well as altered levels of creatine kinase, which predict the rate of disease progression [97]. The elevated serum levels of total and LDL cholesterol and increased levels of apolipoprotein B (apoB), apoB/apoA-I, as well as LDL-C/HDL-C ratio on one hand, but reduced glucose tolerance on the other are factors associated with a higher risk of ALS [98]. Other metabolic biomarkers detected in the blood include decreased serum albumin [97] and glutamine [99], as well as increased glutamate—the metabolite of the latter [100]. However, there are contradictory data regarding glutamate levels in response to therapy [101,102]. Furthermore, a reduction in uric acid [103], and an increase in ferritin, the markers of oxidative stress, have also been identified as promising biomarker candidates [104,105].

Urine biomarkers include the previously described 8-OHdG and 15-F(2t)-IsoP. Studies have also found decreased urinary levels of collagen type 4 [106], and glucosylgalactosyl hydroxylysine [107], a collagen metabolite, as well as increased levels of the extracellular domain of neurotrophin receptor p75 [108].

Aside from the identification of body fluid-based biomarkers, other methods such as magnetic resonance imaging (MRI), are also being used as biomarker tools. Together with structural MRI, several other methods, namely diffusion tensor imaging (DTI), a combination of structural MRI and DTI, magnetization transfer imaging, functional MRI, and proton magnetic resonance spectroscopy, are being investigated to provide insight into the central nervous system of ALS patients. Although less common with the majority of the published work in the context of ALS focusing on the brain, secondary effects on peripheral nerves can also be studied using the DTI method [60].

## 5. EVs as Biomarkers in ALS

Extracellular vesicles (EVs) are intriguing biomarkers for neurodegenerative disorders as is ALS, due to their ability to cross the blood–brain barier (BBB) and peripheral availability [109]. They have been isolated from various biofluids; for example, blood, urine, saliva, and CSF. Additionally, their unique molecular information varies significantly depending on the parent cell type and pathophysiological conditions, thus making them prognostic and diagnostic markers of disease pathogenesis [110]. Another advantage of EVs as biomarkers is their ability to store their contents safely for a longer time due to their double membrane which protects the proteins and RNAs from degradation [111]. To date, EVs have been investigated as sources of potential ALS biomarkers with a focus not only on the EV-associated cargo but also on their number and size. Large EVs (also known as exosomes or LEVs) are vesicles with a diameter of 100–1000 nm, shed by budding of the plasma membrane both in physiological but particularly in pathological conditions. Whereas small EVs (also known as microvesicles or SEVs) are vesicles with a diameter of 30–150 nm, which are formed intracellularly and released by exocytosis of multivesicular bodies [112] (Figure 1).

One of the first observations was an increase in the number of leukocyte-derived LEVs in the CSF of ALS patients [113]. Later, this increase was found to slightly correlate with the rate of ALS progression therefore leukocyte-derived LEVs might be presented as a disease progression biomarker [114]. Regarding the size distribution of EVs, an increase in the mean size of plasma LEVs and SEVs of ALS patients in comparison to the healthy controls was reported. However, no numerical variation was observed between the two groups [115]. No significant difference in the EV numbers between ALS patients and controls was also confirmed by Thompson et al. using CSF samples [116], as well as by Lo et al., who studied the serum samples [117]. However, Lo et al. detected a decrease in the mean size of ALS patients’ serum EVs compared to controls. This, however, disagrees with the findings of Sproviero et al., who found increased LEV and SEV mean size in the plasma of sALS patients [115]. The difference might stem from different isolation and size determination protocols, as well as the influence of various biological factors [117].

### 5.1. EV Protein Cargo in the CSF of ALS Patients

The potential biomarker roles of EV cargo have been studied in the CSF of ALS patients, despite some of its disadvantages such as invasive collection. Misfolded SOD1, as a hallmark of ALS disease, has not yet been found in CSF-derived EVs [118], despite its proven presence in the CSF of ALS patients [119]. However, C-terminal but also full-length TDP-43 fragments were found in EVs isolated from ALS/FTD patients’ CSF samples [120]. Proteomic analysis of exosome-enriched fractions isolated from CSF of sALS patients revealed an increase in INHAT repressor (NIR) which was also found to localize in exosome-like vesicles of CSF as confirmed by immunoelectron microscopy. Furthermore, the NIR subcellular distribution in motor neurons of ALS patients was found to be altered, showing that NIR might be involved in disease pathogenesis [121]. The same authors found three increased proteins (nucleolar complex protein 2 homolog (NOC2L), programmed cell death 6-interacting protein (PDCD6IP), and versican core protein (VCAN)) and 11 decreased proteins (alpha-1-antichymotrypsin (SERPINA3), receptor-type tyrosine-protein phosphatase zeta (PTPRZ1), complement C1q subcomponent subunit C (C1QC), coiled-coil domain-containing protein 19, mitochondrial (CCDC19), myosin light chain 6B (MYL6B), macrophage receptor (MARCO), IgG Fc-binding protein (FCGBP), folate receptor alpha (FOLR1), reelin (RELN), complement factor B (CFB), and charged multivesicular body protein 4a (CHMP4A)) in CSF-derived exosomes of ALS patients compared to controls. Thompson and co-workers performed the EV proteome analysis and found differentially abundant BLMH protein in ALS samples, suggesting its potential role in ALS pathogenesis [116] (Table 1).

### 5.2. EV miRNA Cargo in the CSF of ALS Patients

Apart from proteins, preliminary evidence featuring exosomal miR-124-3p, derived from the CSF, was presented by Yelick et al. Their results showed miR-124-3p levels significantly correlated with disease severity, indicated by the ALSFRS-R score, which implies miR-124-3p might serve as a prognostic biomarker for ALS [122] (Table 2).

### 5.3. EV mRNA Cargo in the CSF of ALS Patients

Otake et al. demonstrated the potential of exosome-derived mRNAs in CSF as ALS diagnostic biomarkers. They described a promising panel of differentially expressed mRNAs, with CUEDC2 representing the most outstanding exosomal mRNA, which was found to be increased in ALS patients’ CSF [123] (Table 1).

### 5.4. EV Protein Cargo in the Blood of ALS Patients

Blood seems to be a more promising biomarker source because its collection is minimally invasive; however, the biofluid provides a variety of information. In 2005, Caby et al. published the first evidence for the presence of exosomes in vivo, in the blood [124]. They were thought to largely derive from erythrocytes and platelets, but even from endothelial cells and leukocytes in some amount [125,126]. As ALS is a multisystem disease, blood-derived EVs may present information regarding early pathological events [127]. Due to the abundance of blood proteins, it can be difficult to identify the EV proteome; however, specificity can be increased using immunopurification. In particular, neuronal or astrocytic surface markers were described for the isolation of neuronal or astrocytic EVs, respectively [128]. Studies describe LCAM1 immunoprecipitation for the isolation of (presumably) neuron-specific EVs from the plasma [128,129]. On the other hand, a recent study advises against using L1CAM as a marker of neuron-derived EVs (NDEVs), because it may not be associated with EVs in the plasma or CSF [130]. Similarly, biotinylated glutamine aspartate transporter (ACSA-1) antibody was used for the isolation of astrocyte-derived EVs by Chen et al. [131].

Regarding the protein content of blood-derived EVs, LEVs isolated from the plasma of sALS patients were found enriched with SOD1, TDP-43, phosphorylated TDP-43, and FUS. SEVs on the other hand showed no change in their protein cargo, suggesting LEVs as the main carriers of toxic proteins in a prion-like propagation of ALS [115]. Pasetto et al. reported significantly elevated levels of hyperphosphorylated TDP-43 in the plasma EVs [132]. Surprisingly, immunogold TEM revealed no evidence of the phosphorylated protein within the EVs. This suggests that phosphorylated TDP-43 is not located intravascularly, but rather has an affinity for the EV “protein corona.” The latter is represented by various plasma proteins that are either bound to the EV membrane or the nanomaterials which come in contact with biofluids. However, EV-derived TDP-43 was not considered a promising ALS biomarker candidate because it could not differentiate between patients and controls [132]. In a study by Chen et al., baseline plasma-derived EVs from ALS patients and follow-up samples at 1, 3, 6, and 12 months were analyzed. As the disease progressed, EV-derived TDP-43 levels increased, most noticeably at the 3- and 6-month follow-ups. Therefore, they could be employed as a potential biomarker for the clinical follow-up of the disease [133]. With analyses of biofluid-derived EVs, new proteins potentially associated with ALS pathogenesis have been retrieved. For instance, Chen et al. reported an increase in IL-6 found in exosomes derived from astrocytes that were isolated from the plasma of sALS patients. Additionally, IL-6 levels were found to positively correlate with ALS progression; however, IL-6 is unspecific to ALS because its concentration was found elevated in other NDs as well. Nevertheless, it is a sufficient marker for neuroinflammation and disease progression [131]. Furthermore, heat shock protein 90 (HSP90) levels were found significantly decreased in plasma-derived EVs of ALS patients, which is useful for differentiation from other motor neuron-related conditions [132]. Peptidylprolyl isomerase A (PPIA), a chaperone protein and regulator of TDP-43 trafficking and function, is significantly lower in plasma-derived EVs of fast-progressing ALS patients compared to the slow-progressing form of the disease, suggesting it may serve as a predictor of ALS disease progression [132]. Moreover, Zhou et al. found a 5.3-fold higher level of coronin-1a (CORO1A) in the EVs isolated from the ALS patients’ plasma compared to the control samples. Their study also showed an increase in CORO1A levels with ALS progression thus identifying EV-derived CORO1A as a potential biomarker [134]. Neurofilaments are considered unspecific, but are the most promising biomarkers for ALS diagnosis and progression, nevertheless. They were, however, not detected in the plasma-derived EVs and so are presumably released to the biofluids in a free-form [132] (Table 1).

### 5.5. EV miRNA Cargo in the Blood of ALS Patients

In addition to proteins, blood EVs have also been examined for the miRNAs they carry. Indeed, under ND conditions, the miRNA content of the EVs is subject to modifications. Additionally, inside the EVs, miRNA can cross the BBB and be readily detected by RNA sequencing or microarray analysis. Therefore, distinct miRNAs may serve as biomarkers [135]. In Table 2, we collected results of recent studies on the miRNA content of the EVs isolated from the serum and plasma of patients with ALS. One of the first studies was published by Xu et al. who found downregulated expression of miR-27a-3p in serum-derived EVs from ALS patients compared to controls. miR-27a-3p was defined as a potential reference for the disease diagnosis [136]. A study by Katsu et al. was the first to describe miRNA expression profiles in neuronal EVs, isolated from ALS patients’ plasma, obtained by microarray analysis. LCAM1 antibodies were used to selectively isolate neuronal exosomes; however, the sample size was small with five patients and five controls. They reported 30 differentially regulated exosomal miRNAs in the plasma of ALS patients. To elaborate, 13 miRNAs were elevated whereas 17 miRNAs were reduced. As revealed by gene ontology analysis, biological processes implicated in those dysregulated miRNAs are part of synaptic vesicle-related pathways. When comparing the miRNAs to those in the formalin-fixed paraffin-embedded (FFPE) samples from the ALS motor cortex, miR-24-3p, miR-1268a, miR-3911, and miR-4646-5p were dysregulated in the same manner. The study suggests miRNAs within NDEVs hold clinical advantages as potential ALS biomarkers because they mirror the miRNA changes in the brain, but validation on a larger sample size is yet to be performed [129]. A follow-up study by Banack et al. also used the LCAM1 immunoprecipitation method but a larger sample size—a total of 40 patients and controls. Next-generation sequencing (NGS) following qPCR analysis was conducted with the following results: five miRNAs were upregulated, and three miRNAs were downregulated in the neuronal-derived EVs, isolated from ALS patients relative to healthy controls. The study aimed to identify miRNA fingerprints from neural-enriched EVs which would shorten the time to ALS diagnosis [137]. However, studies by Katsu et al. and Banack et al. revealed no overlapping miRNA sequences. Moreover, Pregnolato et al. published the data stating studies performed by Katsu et al. and Banack et al. lack the adjustment for multiple testing, resulting in an increased rate of false positive conclusions. Their results on a rather small sample size (n = 7) using an RT-qPCR analysis showed no statistically significant difference in the expression of miRNA in serum-derived exosomes among the ALS and control group. As they stated, the divergence may result from the sample variability, different vesicle isolation protocols, and different chips used for quantification of miRNA expression profile, as well as the difference in normalization procedures. Nevertheless, miRNAs with the highest content in exosomes were found to be involved in ALS-related pathways, namely ‘‘neurotrophin TRK receptor signaling,’’ ‘‘stress response,’’ and ‘‘modification process of cellular protein,’’ suggesting they take part in neurodegeneration [138]. The study by Saucier et al. revealed five elevated and 22 reduced levels of miRNAs in plasma-derived exosomes of ALS patients. Furthermore, miR-15a-5p and miR-193-5p were described with either diagnostic potential or associated with ALS progression, respectively [139]. Lo et al. contributed to the ALS biomarker research with the demonstration that the circulating EVs cargo reflects the state of CNS in ALS. In particular, miR-450a-2-3p, miR-587, and miR-298 levels parallelly differed in the CNS tissue and serum of ALS patients, therefore, supporting the potential of EVs as diagnostic and prognostic biomarkers. They also discovered 11 elevated miRNAs and five reduced miRNAs in ALS samples compared to controls. Two of the miRNAs were overlapping with the profiles of ALS EV-associated deregulated miRNAs previously described in the literature [117]. Banack et al. and Saucier et al. observed reduced levels of miR-4454 in ALS plasma and serum EVs; on the contrary, the same miRNA was found elevated in ALS serum-derived EVs by Lo et al. [137,139]. Similarly, Saucier et al. reported reduced miR-127-3p levels, but elevated levels were reported by Lo et al. [139]. Perhaps the miRNA profiles of EVs are alternating through the ALS course; therefore, future longitudinal studies with larger sample sizes and serial sampling are needed [117]. Another study by Sproviero et al. identified 109 differentially expressed miRNAs in SEVs and 197 in LEVs of ALS patients compared to healthy controls. Among them, 45 upregulated and 22 downregulated miRNAs overlapped between SEVs and LEVs. They found 13 common miRNAs among ALS, AD, PD, and FTD patients. To elaborate, two miRNAs were upregulated (miR-133a-3, miR-4451) and four downregulated (miR-543, miR-6889-5p, miR-4781-3p, miR-323-3p) in SEVs. Furthermore, one miRNA (miR-4433b-5p was upregulated and six were downregulated (miR-1262, miR-3152-3p, miR-7856-5p, miR-365a, miR-6068, miR-767-30) in LEVs. They showed different sorting of miRNA in LEVs and SEVs as well as distinguished ALS from other NDs by examining SEV and LEV-derived miRNAs [140]. Taken together, plasma-derived LEVs and SEVs differ significantly not only in their size, markers, and protein content but also in miRNA content as observed in several studies by Sproviero et al. [114,115,140]. Banack et al. identified miRNA biomarkers that may not only statistically differentiate ALS patients from healthy controls but can also be repeatably measured [4]. They reproduced their previous study with some adjustments, such as a larger sample size, less strict blood collection protocols, removal of the criteria for ALS disease stage, and diagnosis determination by multiple neurologists [137]. The results identified five out of eight originally determined miRNAs, with consistent directionality of dysregulation. The authors believe the presented miRNA fingerprint is robust and worthy of the following studies for clinical use [4] (Table 2).

### 5.6. EV mRNA and lncRNA Cargo in the Blood of ALS Patients

Apart from the proteins and miRNAs mentioned so far, EVs can carry other non-coding RNAs, for instance, tRNA, rRNA, lncRNA, and piwi-interacting RNA (pi-RNA), as well as coding RNA (i.e., mRNA) [141]. Another study by Sproviero et al. provided the first evidence of mRNAs and lncRNAs in the plasma EVs of patients with ALS. When comparing deregulated mRNAs and lncRNAs transported by SEVs and LEVs, only the mRNAs were found to overlap partially. A higher number of differentially expressed mRNAs were found in SEVs compared to LEVs; however, there was no significant difference regarding lncRNAs in SEVs and LEVs. Furthermore, differentially expressed mRNAs in SEVs and LEVs were found to differ from controls, which was also observed for lncRNAs in both SEVs and LEVs. A total of 15 mRNAs are shared, meaning that a specific common signature can be described for the two EV subpopulations [142] (Table 1).

### 5.7. EV Lipid and Metabolite Cargo in the Blood of ALS Patients

Aside from protein and nucleic acid EV cargo, a study by Morasso et al. was the first to give information about different lipid signatures in the EVs of sALS patients. Using Raman spectroscopy, they showed different biochemical profiles of LEVs in sALS patients’ plasma compared to controls. Furthermore, they found LEVs of ALS patients were especially rich in lipids and deficient in phenylalanine which perhaps mirrors the differences in LEV protein content. Lipids found on SEVs were much less marked than the ones on LEVs. Lastly, the study suggests LEVs as promising biomarkers for ALS diagnosis and provides evidence for the role of lipids and phenylalanine metabolism in LEVs of sALS patients with potential involvement in ALS pathogenesis [143] (Table 1).

**Table 1 genes-14-00325-t001:** Other than miRNA potential EV biomarkers for ALS.

Biofluid	EV Subpopulation	ALS Patients (n)	Controls (n)	Biomarker Identified	Results in ALS Patients Compared to Controls	Author, Year
**Size distribution of EVs**						
CSF	Leukocyte-derived EVs	40	28 AD 36 HC	Number	Increased	Sproviero et al. (2019) [114]
CSF	EVs	20	9 HC	Number	No variation	Thompson et al. (2020) [116]
Plasma	LEVs and SEVs	30	20 HC	Number Mean size	No variation Increased	Sproviero et al. (2018) [115]
Serum	EVs	15	16 HC	Number Mean size	No variation Decreased	Lo et al. (2021) [117]
**Protein biomarkers**						
CSF	SEVs	18 (ALS) 8 (ALS/FTD)	15 HC	C-terminal, full-length TDP-43 fragments	Increased	Ding et al. (2015) [120]
CSF	SEVs	3	3 iNPH	NIR, NOC2L, PDCD6IP, VCAN	Increased	Hayashi et al. (2020) [121]
				SERPINA3, PTPRZ1, C1QC, CCDC19, MYL6B, MARCO, FCGBP, FOLR1, RELN, CFB, CHMP4A	Decreased	
CSF		20	9 HC	BLMH	Increased	Thompson et al. (2020) [116]
Plasma	LEVs	30	20 HC	SOD1, TDP-43, p-TDP-43, FUS	Increased	Sproviero et al. (2018) [115]
Plasma	LEVs	40	39 HC	IL-6	Increased	Chen et al. (2019) [131]
Plasma	SEVs	18	/	TDP-43	Increasing overtime	Chen et al. (2020) [133]
Plasma	EVs	106	36 HC 28 MD 32 SBMA	Hyperphosphorylated TDP-43	Increased	Pasetto et al. (2021) [132]
				HSP90, PPIA	Decreased	
Plasma	EVs	3	3 HC	CORO1A	Increased	Zhou et al. (2022) [134]
**RNA biomarkers**						
CSF	LEVs	4	4 HC	CUEDC2 (mRNA)	Increased	Otake et al. (2019) [123]
Plasma	LEVs and SEVs	6	6 HC 6 AD 9 PD 9 FTD	CUL3, HIST3H2A, SRSF11, RAB18, SRSF1, DDX5, TRPS1, NFIA, ARIH1, DHX15, NFIX, TUBB1, NF1, NFIB, HUWE1 (mRNA)	Differed from controls	Sproviero et al. (2022) [142]
**Lipid and metabolite signatures**						
Plasma	LEVs	20	20	Lipids Phenylalanine	Increased Decreased	Morasso et al. (2020) [143]

HC, healthy controls; AD, Alzheimer’s disease; iNPH, idiopathic normal-pressure hydrocephalus; MD, muscular dystrophy; SBMA spinal and bulbar muscular atrophy; PD, Parkinson’s disease; FTD, frontotemporal dementia.

**Table 2 genes-14-00325-t002:** Potential EV miRNA biomarkers for ALS.

Biofluid	EV Isolation Method	Analysis	ALS Patients (n)	Controls (n)	miRNA Identified	Results in ALS Patients Compared to Controls	Author, Year
Serum	ExoQuick/ ExoEasy Kit	qRT-PCR	10	20 HC	miR-27a-3p	1 reduced	Xu et al. (2018) [136]
Plasma	Polyethylene-glycol/ L1CAM immuno-precipitation	Microarray	5	5 HC	miR-4736, miR-4700-5p, miR-1207-5p, miR-4739, miR-4505, miR-24-3p, miR-149-3p, miR-4484, miR-4688, miR-4298, miR-939-5p, miR-371a-5p, miR-3619-3p	13 elevated	Katsu et al. (2019) [129]
					miR-1268a, miR-2861, miR-4508, miR-4507, miR-3176, miR-4745-5p, miR-3911, miR-3605-5p, miR-150-3p, miR-3940-3p, miR-4646-5p, miR-4687-5p, miR-4788,miR-4674, miR-1913, miR-634, miR-3177-3p	17 reduced	
Plasma	Nv96 peptide	NGS	14	12 HC	miR-532-3p, miR-144-3p, miR-15a-5p, miR-363-3p, miR-183-5p	5 elevated	Saucier et al. (2019) [139]
					**miR-4454**, miR-9-1-5p, miR-9-2-5p, miR-9-3-5p, miR-338-3p, miR-100-5p, miR7977, miR-1246, miR-664a-5p, miR-7641-1 miR-1290, miR-4286, miR-181a-1-5p, miR-181a-2-5p, miR-181b-1-5p, miR-181b-2-5p, miR1260b, miR-199a-1-3p, miR-199b-3p, miR-199a-2-3p, **miR-127-3p**, let-7c-5p	22 reduced	
Serum	SPB ExoQuickl/L1CAM immuno-precipitation	NGS	20	20	**miR-146a-5p**, miR-199a-3p, miR-151a-3p, **miR-151a-5p**, miR-199a-5p	5 elevated	Banack et al. (2020) [137]
					**miR-4454**, **miR-10b-5p**, **miR-29b-3p**	3 reduced	
CSF	Total Exosome Isolation Reagent-mediated precipitation	qRT-PCR	14	9 DC 9 HC	**miR-124-3p**	1 elevated	Yelick et al. (2020) [122]
Serum	ExoChip	Nano String	14	8	miR-520f-3p, **miR-4454**, miR-7975, miR-450a-2-3p, miR-1268b, miR-26a-5p, miR-1255a, miR-342-3p, **miR-127-3p**, miR-551b-3p, miR-1262	11 elevated	Lo et al. (2021) [117]
					miR-877-5p, miR-298, miR-766-3p, miR-587, miR-1254	5 reduced	
Plasma	Centrifugation/Ultra-centrifugation	NGS	6	6 HC 6 AD 9 PD 9 FTD	miRNA-8089, miRNA-196a-5p, miRNA-3152-3p*, miRNA-607, miRNA-3607-3p, miRNA-6825-3p, miRNA-7106-5p, miRNA-3976, miRNA-4492, miRNA-200a-3p, miRNA-205-5p, miRNA-6858-3p, miRNA-1273c, miRNA-6888-3p, miRNA-4302, miRNA-4634, miRNA-182-3p, miRNA-3160-3p, miRNA-1-3p, miRNA-200a-5p, miRNA-7704, miRNA-210-3p, miRNA-31-5p, miRNA-133a-3p*, miRNA-34c-5p, miRNA-455-5p, miRNA-6842-5p, miRNA-3619-3p, miRNA-4279, miRNA-4508, miRNA-1469, miRNA-141-3p, miRNA-542-3p, miRNA-615-3p, miRNA-200c-3p, miRNA-4451*, miRNA-18a-5p, miRNA-200b-3p, miRNA-184, miRNA-9-5p, miRNA-7c-5p, miRNA-6746-5p, miRNA-3195, miRNA-206, miRNA-6068	45 elevated	Sproviero et al. (2021) [140]
					miRNA-493-3p, 409-3p, miRNA-323b-3p*, miRNA-6073, miRNA-432-5p, miRNA-134-5p, miRNA-330-3p, miRNA-625-3p, miRNA-4446-3p, miRNA-148b-3p, miRNA-370-3p, miRNA-584-5p, miRNA-224-5p, miRNA-381-3p, miRNA-199a-5p, miRNA-654-3p, miRNA-335-3p, miRNA-543*, miRNA-4433b-5p*, miRNA-130b-5p, miRNA-4286, miRNA-382-5p	22 reduced	
Plasma	SPB ExoQuickl/L1CAM immuno-precipitation	qRT-PCR	50	50	**miR-151a-5p,** **miR-146a-5p**	2 elevated	Banack et al. (2022) [4]
					**miR-4454,** **miR-10b-5p,** **miR-29b-3p**	3 reduced	

## 6. Discussion

ALS is a fatal ND for which there is currently no cure despite great efforts. Because the initial symptoms of ALS overlap with the symptoms of several other NDs and the diagnosis is based mainly on clinical examination, it may take up to a year to confirm the disease [144]. Diagnosis and detection of ALS is a major challenge for researchers and clinicians. Genetic testing for the most common mutations such as C9ORF72, SOD1, TDP43, FUS, and TBK1 can contribute to earlier diagnosis, but mutations in these genes explain only a few percent of all ALS cases. Therefore, it is critical to identify and validate biomarkers that would shorten the diagnostic process and improve diagnostic accuracy [145]. ALS biomarkers also play an important role in prognosis, as they can indicate the onset of the disease in the early stages before the clinical symptoms appear [1]. Although there are a large number of research data in the field of ALS biomarkers, few have been validated. The main reason for this are: variable and non-standardized methodology, small patient cohorts, and a lack of longitudinal studies [60].

In this review, we compiled current information on biomarker identification, focusing in particular on EVs as biomarkers for ALS. EVs are small membrane-bound vesicles released by cells into the extracellular space and represent one of the forms of intercellular communication. Their main function is to exchange information with other cells through their cargo of proteins, lipids, metabolites, and nucleic acids, which can directly affect the microenvironment. This form of intercellular communication can be deregulated and potentially contribute to disease development. The release of EVs from a cell can provide valuable insight into the intracellular processes within the cell, which may contribute to the understanding of disease pathophysiology and the development of diagnostic and therapeutic strategies [146].

The use of EVs may be the key to accessing pathologically relevant tissues for ALS because EVs overcome the BBB and can be easily isolated from biological fluids. The number, size, and content of EVs containing different types of signals can provide clues to the disease, its stage, and prognosis [109]. EVs were isolated from the CSF, blood, and urine of ALS patients and control subjects and compared in terms of their size and content. Regarding the average size of serum EVs from ALS patients compared to control subjects, studies have been inconsistent, which could be due to different methods or different biological influences [115,117].

Proteomic studies of CSF-derived EV cargo revealed that several proteins such as TDP-43, NIR, NOC2L, PDCD6IP, VCAN, and BLHM were increased and SERPINA3, PTPRZ1, C1QC, CCDC19, MYL6B, MARCO, FCGBP, FOLR1, RELN, CFB, and CHMP4A were decreased, indicating their potential role in ALS pathogenesis [116,121].

RNA cargo of EVs in CFS of ALS patients identified miR-124-3p as a potential prognostic ALS biomarker correlating with disease severity [122], and CUEDC2 mRNA as the most elevated coding RNA was proposed as a potential disease biomarker for ALS [123] (see Table 1).

Biomarkers should be minimally invasive and easily accessible in living patients; thus, blood seems to be a more promising biomarker source than CSF [147]. LEVs, but not SEVs, isolated from the plasma of sporadic ALS patients were enriched in the toxic ALS-related proteins SOD1, TDP-43, phospho-TDP-43, and FUS compared with healthy controls [115]. Moreover, the EV-derived TDP-43 levels increased during the course of the disease and could be considered as a potential biomarker for clinical follow-up of the disease [133]. Both HSP90 and PPIA levels were found to be decreased, with HSP90 having greater diagnostic potential and PPIA having greater prognostic potential [132]. Increased levels of CORO1A from EVs correlated with the progression of ALS and revealed CORO1A as a potential biomarker [134]. To specify the EV cargo relevant to the disease, astrocyte EVs were isolated from plasma by immunopurification. The elevated levels of IL -6 detected could be a sufficient marker of neuroinflammation [131].

Although EV mRNA content differs between LEVs and SEVs, a specific common signature of 15 differentially expressed mRNAs shared by both SEVs and LEVs has been detected in the serum of ALS patients compared to healthy controls and patients with other NDs [142].

In addition to proteins and RNAs, lipids and metabolites constitute EVs cargo as well [143]. Interestingly, only LEVs but not SEVs in the plasma of sALS patients have a different biochemical profile compared to healthy controls. They were particularly rich in lipids and poor in the aromatic amino acid phenylalanine, a promising biomarker for the diagnosis of ALS. The death of motor neurons in ALS could disrupt the lipid organization of the presynaptic membrane, affecting membrane trafficking [148] and the lipid cargo of LEVs [143].

In most studies, EVs were analyzed for their miRNA content. As shown in Table 2, different EV isolation and miRNA analysis methods were used, resulting in different miRNA profiles. EVs were isolated from either plasma or serum, and in some studies, immunopurification was also used to isolate (presumably) neuron-specific EVs from plasma or serum [4,129,137]. miR-4454 and miR-127-3p were deregulated in more than one study, but in opposing directions [117,137,139]. The predicted targets of miR-4454 are closely related to neurogenesis, synapse formation, and motor neuron integrity [149,150]. miR-127-3p targets are associated with ALS and have known effects on axon guidance, histone acetylation, and intracellular motor proteins [151]. miR-127-3p is also part of an exosomal diagnostic biomarker panel for multiple sclerosis [152]. Five miRNAs (miR-151a-5p, miR-146a-5p upregulated and miR-4454, miR-10b-5p, miR-29b-3p downregulated) repeatedly discriminated between sALS patients and healthy controls under robust conditions and were proposed as a fingerprint for EV miRNA [4].

Taken together, the results discussed above represent an important contribution to the identification of EV biomarker candidates for ALS by identifying disease-relevant factors in plasma and/or serum EVs. It is important to distinguish between different subtypes of EVs, as reducing the heterogeneity of EV samples will greatly improve diagnostic interpretation. The different molecular profiles of LEVs and SEVs confirmed the distinct functional roles of LEVs and SEVs in the plasma of patients with ALS, as described for the dimension of EVs, protein, miRNA, and lipid cargo, thus supporting the decision to analyze them independently in a case-control study [114,115,142,143].

In a GWAS, we recently confirmed that perturbations of vesicle-mediated transport and autophagy play a role in ALS [153]. Mandrioli et al. also found that the close relationship between SEV release and autophagy is critical for the development of ALS [154]. In addition, proteins associated with ALS, such as optineurin, valosin-containing protein, ubiquilin-2, and p62, contribute to recruiting proteins to the autophagosome for degradation. Heat shock protein B8 (HSPB8) recognizes and promotes an autophagy-mediated removal of misfolded mutant SOD1 and TDP-43 fragments, and aggregated dipeptide species produced by ALS motor neurons in C9ORF72-related diseases from ALS motor neurons [154]. Moreover, the transcriptional profile of mutant SOD1-expressing mice showed differential expression of exosomal and lysosomal genes, suggesting that these processes are affected together in ALS [155]. We also demonstrated a causal role of cholesterol, which may be associated with impaired autophagy that promotes susceptibility to ALS [153]. Hyperlipidemia as a causal risk factor for ALS was also recognized in a previous GWAS [156].

The enhanced deregulation of protein, RNA, lipid, and metabolite EV cargo referenced in this review might be due to the exosomal pathway associated with dysfunctional autophagy and the endolysosomal pathway in ALS. Inhibition of lysosomes has been reported to increase the number of amphisomes in the neuronal cell line and increase the level of autophagy-related proteins in exosome-like EVs, supporting a model in which cells use exosome secretion to remove protein aggregates during lysosomal or autophagic dysfunction [157,158]. Because lysosomes also mediate lipid removal, lipid storage may be impaired [159], and the exosomal pathway associated with autophagy and the endolysosomal pathway may also lead to the deregulation of RNA observed in SEVs [160].

In our recent study, we provided evidence for the cell-autonomous initiation of ALS in glutamatergic neurons [153]. Although several studies have used LCAM1 immunoprecipitation to isolate (presumably) neuron-specific EVs from plasma [128,129], a recent study has shown that L1CAM is not associated with EVs in human plasma or CSF and therefore advises against its use as a marker in neuronal EV isolation protocols [159]. Thus, the discovery of more specific markers and the development of more innovative separation methods for the isolation of circulating EVs from the brain, as well as the identification of their origin in specific types of brain cells, remains a challenge [161]. Most of the studies reported so far on the identification of circulation EV biomarkers for ALS are relatively small in scope. Therefore, further studies on larger patient and control cohorts are needed to clarify the power, sensitivity, and specificity of specific EV contents before the EV-based diagnosis of ALS becomes feasible.

## 7. Conclusions

Thus, isolation and content analysis of neuron-derived EVs from biological fluids, especially blood, is probably the most appropriate approach to identify relevant biomarkers for ALS diagnosis and progression. However, in this regard, there is still a lack of consensus on protocols for standardization of the isolation and analysis of EVs. Although there are still some technical challenges and learning needs that must be addressed to fully exploit the potential of EVs, we believe that in the near future, EVs may become clinically useful biomarkers and a source of information about disease pathogenesis and may even provide an alternative therapy for currently untreatable diseases.

## Figures and Tables

**Figure 1 genes-14-00325-f001:**
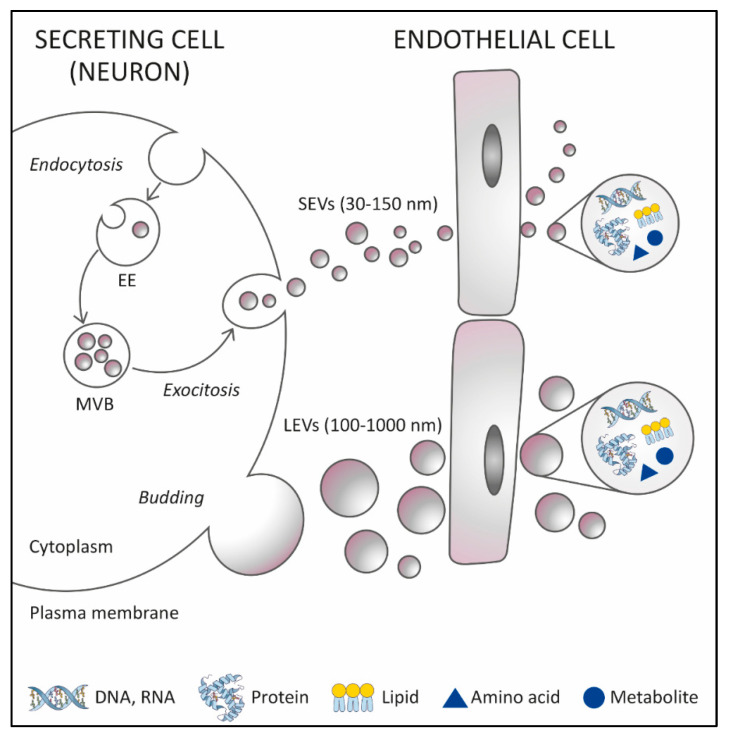
Formation of extracellular vesicles (EVs) and transport across the blood–brain barrier (BBB). Large EVs (LEVs) and small EVs (SEVs) are two subtypes of EVs, that can be distinguished by their size and biogenesis. LEVs are shed by budding of the plasma membrane, whereas SEVs are formed intracellularly and released by exocytosis of multivesicular bodies (MVBs). EVs can cross the BBB and thus enable various components, specific to secreting cells, including nucleic acids, proteins, amino acids, lipids, and metabolites in the EVs to be transported from the central nervous system (CNS) to peripheral biofluids. EE, early endosome.

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
