# Peer review of "Extracellular Vesicles as Potential Biomarkers in Amyotrophic Lateral Sclerosis"

_genes, 2023, doi:10.3390/genes14020325_

Round 1

Reviewer 1 Report

The manuscript by Barbo and Ravnik-Glavač represents an interesting body of research evidence in the ALS biomarkers especially in extracellular vesicles field of study. The manuscript well explained all ALS biomarkers in different fluid of body in literature and compare them in two groups of ALS and non-ALS. However, some revisions must be done in this manuscript.

L119-227. The authors well explained the biomarkers related to ALS, but as it is obvious, most of these markers change in so many other diseases. The authors must clarify which of these biomarkers are exclusive for ALS or have more sensitive for it.

L242. Figure 1 has lack of quality and also the cell membrane should be better draw. It appears it was draw by hand.

However, authors well wrote about the biomarkers of the ALS; but the discussion has no sufficient data about the manuscript. The discussion section must have an overall overview of whole study and moreover it must include key results of the study and some discuss and ague about it.

Discussion section must have references. Please insert reference(s) for each sentence.

Please separate conclusion section from discussion and elaborate the conclusion of the current study in detail.

Reviewer 2 Report

The figure can be improved in the review.

A table containing molecules other than miRNA can also be incorporated in the review.

Reviewer 3 Report

The review article "Extracellular vesicles as potential biomarkers in amyotrophic lateral sclerosis" has summarized the use of Extracellular vesicles in the prognosis of amyotrophic lateral sclerosis. The ALS progression is challenging to diagnose and employing EVs as biomarkers have huge potential to detect it early.  The review is interesting and has been discussing an important topic in neurodegenerative diseases. Below are a few comments that the authors might find interesting. 

1) Authors have missed a few most important articles that are highly cited on similar topics published recently. Thus they are suggested to include those as well.

2) Authors should seriously consider upgrading figure 1 in the review, and also making it more comprehensive. It would be great if more figures were throughout the review.

3) Authors should check the reference formatting for the as required in reference numbers 7, 23, 4754, 87, 89,98& 103.

4) Authors if possible could divide the section "EV as Biomarkers in ALS" into subsections.

5) Authors are needed to add some comments about the future work direction for EVs as biomarkers in ALS. Finally, they should also mention about big hurdles/ technical issues or limitations to such studies in the summary.

Round 2

Reviewer 1 Report

Thank you for considering my suggestions.